# A MEMS-Based High-Fineness Fiber-Optic Fabry–Perot Pressure Sensor for High-Temperature Application

**DOI:** 10.3390/mi13050763

**Published:** 2022-05-12

**Authors:** Suwei Wang, Jun Wang, Wenhao Li, Yangyang Liu, Jiashun Li, Pinggang Jia

**Affiliations:** State Key Laboratory of Dynamic Measurement Technology, North University of China, Taiyuan 030051, China; s1906162@st.nuc.edu.cn (S.W.); b20210625@st.nuc.edu.cn (J.W.); s202106055@st.nuc.edu.cn (W.L.); s2006037@st.nuc.edu.cn (Y.L.); b1806016@st.nuc.edu.cn (J.L.)

**Keywords:** high temperature, pressure sensor, fiber-optic Fabry–Perot, high fineness

## Abstract

In this paper, a high-fineness fiber-optic Fabry–Perot high-temperature pressure sensor, based on MEMS technology, is proposed and experimentally verified. The Faber–Perot cavity of the pressure sensor is formed by the anodic bonding of a sensitive silicon diaphragm and a Pyrex glass; a high-fineness interference signal is obtained by coating the interface surface with a high-reflection film, so as to simplify the signal demodulation system. The experimental results show that the pressure sensitivity of this sensor is 55.468 nm/MPa, and the temperature coefficient is 0.01859 nm/°C at 25~300 °C. The fiber-optic pressure sensor has the following advantages: high fineness, high temperature tolerance, high consistency and simple demodulation, resulting in a wide application prospect in the field of high-temperature pressure testing.

## 1. Introduction

The high-temperature pressure sensor has a wide range of application requirements in the fields of aerospace, oil exploration, nuclear reactors, and others [1,2,3,4,5,6]. For example, the pressure measurement of various aerospace engine pipelines is important for analyzing engine thrust [7]. In order to reduce the cost of oilfield exploitation and ensure safety in the process of oilfield exploitation, it is extremely important to measure pressure parameters downhole in a high-temperature environment [8]. In a nuclear power plant, it is necessary to monitor the pressure parameters of the nuclear reactor in a high-temperature environment to ensure the safety and functionality of the nuclear power plant [9].

Many different types of pressure sensor have been developed in recent years, including piezoresistive [10], piezoelectric [11], capacitive [12] and fiber-optic sensors [13,14,15]. Compared with traditional electrical sensors, fiber-optic sensors have the advantages of small size, high sensitivity, and anti-electromagnetic interference signals [16,17,18,19]. Ma et al. developed a fiber-optic Fabry–Perot pressure sensor based on the microbubble structure, with high sensitivity. However, the proposed sensor can only be made individually; due to the arc discharge technology, the consistency of the sensor is poor and cannot be mass produced [20]. Feng et al. developed a fiber-optic Fabry–Perot pressure sensor, based on MEMS technology, with good consistency, but the sensor requires a complex signal demodulation system [21]. In order to simplify the demodulation system, Ma et al. proposed a high-fineness fiber-optic Fabry–Perot pressure sensor, but the temperature of the sensor cannot exceed 100 °C, due to the use of a fiber collimator [22].

In this paper, a high-fineness fiber-optic Fabry–Perot pressure sensor, based on MEMS technology, is proposed and experimentally verified in a high-temperature environment. The sensor is bonded by a four-layer silicon–glass–silicon–glass wafer to achieve a high consistency of sensitive units. The Fabry–Perot cavity, with high reflectivity, enables the sensor to obtain high-fineness interference signals during the pressure measurement.

## 2. Operating Principle

The structure diagram of the fiber-optic Fabry–Perot pressure sensor, based on MEMS technology, is shown in Figure 1. The sensor is mainly composed of two parts: one is the sensitive unit made by photolithography, etching, sputtering, bonding, and other processes; and the other is fiber-optic welded with the capillary glass tube. The S1 surface of the Fabry–Perot cavity is plated with a high-reflectivity metal film (reflectivity 99%), the S2 surface of the Fabry–Perot cavity is plated with a high-reflectivity optical film (reflectivity 99%), and the S3 surface is plated with a high-transmittance optical film (transmittance 99.8%). In order to reduce the interference introduced by the fiber end face, the reflectivity of the fiber end face is reduced to 0.2% by the coating process.

Due to the two high-reflectivity surfaces of the Fabry–Perot cavity, high-fineness interference peak signals can be obtained by multi-beam interference. The multi-beam interference can be expressed as follows:(1)$$I=\frac{\left(R_{1}+R_{2}-2\sqrt{R_{1}R_{2}}\cos(4\pi nL/\lambda )\right)}{\left(1+R_{1}R_{2}-2\sqrt{R_{1}R_{2}}\cos(4\pi nL/\lambda )\right)}$$
where $R_{1}$ and $R_{2}$ represent the reflectivity of the S1 and S2 surfaces, respectively; n and L are the refractive index of the medium and the Fabry–Perot cavity length, respectively; and λ is the wavelength of the incident light. 

According to Equation (1), the multi-beam interference simulation spectrum of the sensor is shown in Figure 2. It can be observed from Figure 2 that there is only one spike in the free spectral range, and the feature of the interference spectrum will greatly simplify the demodulation system. When the sensitive diaphragm is pressurized, the length of the Fabry–Perot cavity will decrease correspondingly. The change in cavity length leads to a shift in the position of the interference peak. Therefore, according to the shift of the interference peak, the Fabry–Perot cavity length can be calculated.

In designing the sensitive diaphragm, the small deflection theory is adopted to calculate the deformation variable of the diaphragm under different pressure values. The actual length of the Fabry–Perot cavity under an applied pressure can be expressed as follows:(2)$$\Delta L=\frac{3\left(1-\mu^{2}\right)\cdot R^{4}\cdot P}{16Eh^{3}}$$
where E and μ are the Young’s modulus and Poisson’s ratio of the sensitive silicon diaphragm, respectively; h and R are the thickness and effective radius of the sensitive silicon diaphragm, respectively; P is the pressure on the sensitive diaphragm; and ΔL is the deformation variable of the sensitive diaphragm under different pressures. The detailed parameters of the fiber-optic Fabry–Perot pressure sensor are shown in Table 1.

According to Equation (2), the theoretical value of the full-scale deformation variable of the sensitive diaphragm at room temperature is 246.31 nm. The corresponding diaphragm variable has a peak shift of 31.519 nm on the interference spectrum, so the theoretical sensitivity of the sensor is 63.038 nm/MPa.

The length of the Fabry–Perot cavity will change due to the thermal expansion of the material. The thermal expansion of the Fabry–Perot cavity is divided into three parts: the first part is the thermal expansion of the sensitive silicon diaphragm; the second part is the thermal expansion of the unbonded areas on the glass backplane; and the third part is the thermal expansion of the sidewall of the sensitive diaphragm, which causes thermally induced bending of the sensitive silicon diaphragm. The thermal expansion of the Fabry–Perot cavity, according to the COMSOL simulation software, is shown in Figure 3a. It can be observed that the relative displacement of the center of the two surfaces in the Fabry–Perot cavity can reach 40.69 nm at 300 °C. The peak shift of the interference spectrum of the Fabry–Perot cavity, by MATLAB simulation, is shown in Figure 3b. It is evident that the increase in cavity length, caused by thermal expansion, can cause a peak shift of 5.1 nm on the interference spectrum. The theoretical temperature coefficient of the sensor is 0.0185 nm/°C at 25~300 °C. Figure 3 shows the thermal simulation results of the Fabry–Perot cavity.

## 3. Sensor Manufacturing

The manufacturing process of the fiber-optic Fabry–Perot pressure sensor is shown in Figure 4. Firstly, the mask pattern is created using uniform glue and photolithography, and is developed on a four-inch double-polished silicon wafer with a thickness of 300 μm. Then, ICP deep silicon etching is used to etch the silicon wafers masked with photoresist, as shown in Figure 4a. The first two steps are then repeated on the other side of the silicon wafer, as shown in Figure 4b. A high-reflectivity metal film is sputtered on the silicon surface in the Fabry–Perot cavity by a lift-off process, as shown in Figure 4c. A layer of photoresist is covered on both sides of four-inch double-polished glass with a thickness of 300 μm as a mask, optical films are sputtered on both sides of the glass, and the photoresist on the glass is removed, as shown in Figure 4d. The high-reflectivity metal film is composed of Cr and Au, and the thickness of the Cr and Au is in a ratio of 1:5. The optical film is controlled by adjusting the thickness of the specific refractive index material to control the reflectivity and transmittance, and the two types of optical film are made of different thicknesses of stacked Si and SiO_2_.

Then, the silicon wafer with metal film and the glass with optical films are bonded together through an anodic bonding process, to obtain a Fabry–Perot cavity with high reflectivity, as shown in Figure 4e. In order to fix the sensitive unit with the optical fiber, a layer of four-inch double-polished glass with a thickness of 2 mm needs to be fused with capillary glass. Due to the limitations of the process conditions, it is necessary to use through-hole silicon of the same size as the sensitive silicon diaphragm of the connecting layer between the two glass layers, as shown in Figure 4f, and to complete the last anodic bonding process as shown in Figure 4g. Figure 5a shows the mass-produced sensitive units measuring 4×4  mm; the sensitive unit is then welded with the optical fiber and capillary glass tube as shown in Figure 4h. Figure 5b shows the fiber-optic Fabry–Perot sensor.

## 4. Experiment and Results

A high-temperature pressure test platform is developed to test the sensor in a high-temperature environment. The schematic diagram of the experimental setup is shown in Figure 6. The platform consists of a vacuum system, heating system, pneumatic system and water supply system. The sensor test system is mainly composed of the following: a fiber-optic Fabry–Perot pressure sensor; spectrometer; computer; and fiber-optic connector. The high-temperature pressure test platform provides a temperature–pressure composite environment for the sensor. The sensor is placed on the ceramic tray with through holes, and the fiber-optic sensor is connected to the spectrometer via a fiber-optic connector.

The multi-beam spectrum of the sensor at room temperature is shown in Figure 7a. It can be clearly observed in Figure 7a that the wavelength of the multi-beam interference peak is around 1555 nm, and the signal contrast of the interference peak in the free spectral region can reach up to 20 dB, so that other interference surfaces will not affect the interference signal in the Fabry–Perot cavity. The pressure of the furnace body was then increased to 0.5 MPa, and the wavelength of the peak was recorded every 10 min. The output of the sensor under 0.5 MPa pressure is shown in Figure 7b. It can be observed in Figure 7b that the wavelength of the peak remains almost unchanged, and the change in the wavelength does not exceed 0.042 nm during the constant pressure time. Therefore, it is proven that the sensor has good sealing performance during the manufacturing process.

In order to verify the repeatability of the sensor, the pressure of sensor#1 was boosted from 0 MPa to 0.5 MPa, with a step size of 0.05 MPa, and then the pressure was released back to approximately 0 MPa three times. The measured wavelength is shown in Figure 8. It can be observed that there is a small difference in sensitivity, which may be affected by the different vacuum environments. During three pressure tests, the regression coefficient(*R^2^*) between the pressure of the sensor and the corresponding wavelength is greater than 0.99.

In order to verify the nonlinearity of the sensor, the pressure of sensor#1 was boosted from approximately 0 MPa to 0.5 MPa, with a step size of 0.05 MPa, and then the pressure was released back to approximately 0 MPa. The wavelength values of the sensor under the same pressure are almost the same during the boosting and depressing processes, as shown in Figure 9, and the nonlinearity of the boosting and depressing processes are 0.95% F.S. and 0.87% F.S., respectively. The pressure sensitivity of the sensor at room temperature is 55.468 nm/MPa, and the actual value of the pressure sensitivity is slightly smaller than the theoretical value. The main reason for this may be the deviation in the effective radius and the thickness of the sensitive diaphragm from the theoretical parameter in the manufacturing process.

In order to verify the performance of the sensor in a high-temperature environment, sensor#1 was tested from room temperature to high temperature. The relationship between the peak wavelength and the operating temperature is shown in Figure 10. It can be observed that the peak wavelength increased, with the temperature rising due to the high-temperature increases in the Fabry–Perot cavity. The peak shift of the sensor in the temperature range of 25~300 °C is 5.111 nm. The temperature coefficient is 0.01859 nm/°C, which is close to the theoretical results.

The nonlinearity of the fiber-optic Fabry–Perot pressure sensor is tested at 25 °C, 100 °C, 200 °C, and 300 °C, respectively. The applied pressure is changed from approximately 0 MPa to 0.5 MPa, with a step size of 0.05 MPa. The relationship between the peak wavelength and the pressure of the sensor at different temperatures is shown in Figure 11. It can be observed in Figure 11 that the sensitivity of the sensor increased with the temperature, which is determined by the relationship between the Young’s modulus of the material and temperature. When the temperature reaches 300 °C, the sensitivity of the sensor is 57.908 nm/MPa.

## 5. Conclusions

In this paper, a high-fineness fiber-optic Fabry–Perot high-temperature pressure sensor, based on MEMS technology, is proposed and experimentally verified. The Fabry–Perot cavity of the pressure sensor is formed by anodic bonding of a sensitive silicon diaphragm and a Pyrex glass, and a high-fineness signal is obtained by coating the interface surfaces with a high-reflection film, so as to simplify the signal demodulation system. The experimental results show that the pressure sensitivity of the sensor is 55.468 nm/MPa at room temperature, and the temperature coefficient is 0.01859 nm/°C at 25~300 °C. The fiber-optic pressure sensor has the following advantages: high fineness, high temperature tolerance, high consistency, and simple demodulation, demonstrating high prospects in the field of high-temperature pressure testing.

## Figures and Tables

**Figure 1 micromachines-13-00763-f001:**
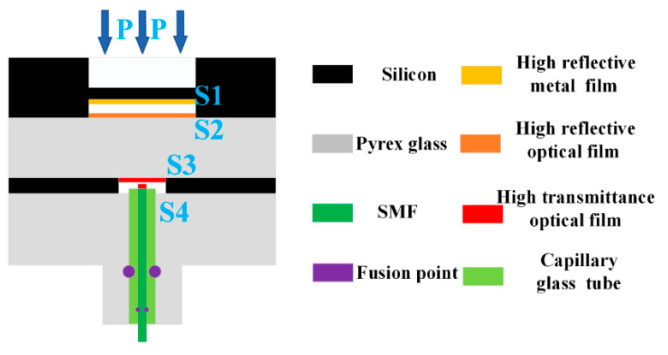
The structure diagram of the proposed fiber-optic Fabry–Perot pressure sensor.

**Figure 2 micromachines-13-00763-f002:**
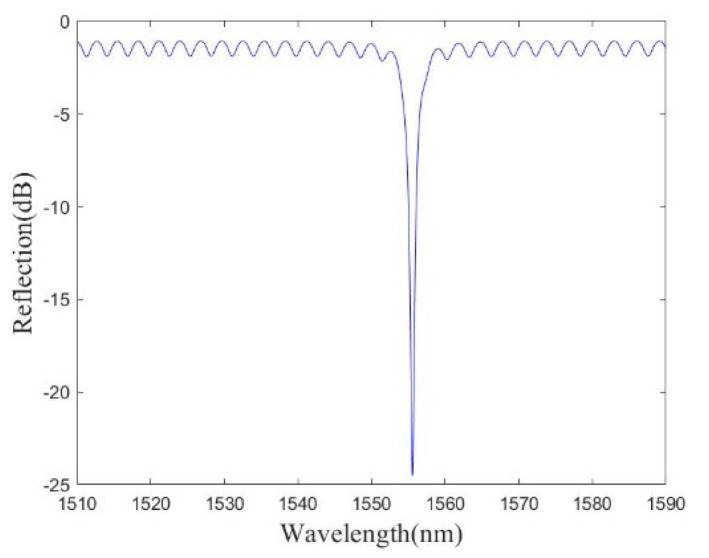
Simulation of multi-beam interference spectrum.

**Figure 3 micromachines-13-00763-f003:**
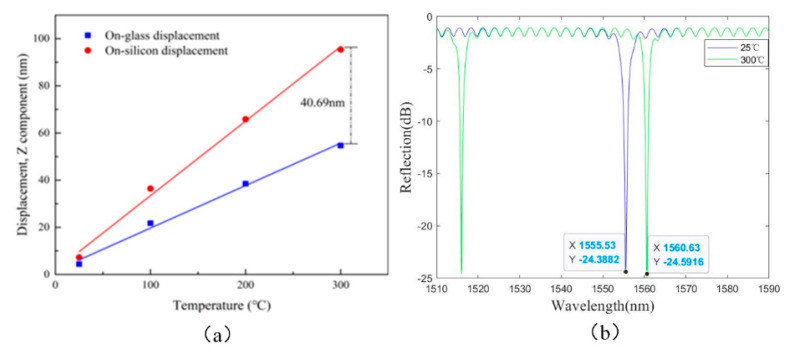
The thermal simulation results of the Fabry–Perot cavity: (**a**) the thermal expansion of the Fabry–Perot cavity, and (**b**) the peak shift of the interference spectrum of the Fabry–Perot cavity.

**Figure 4 micromachines-13-00763-f004:**
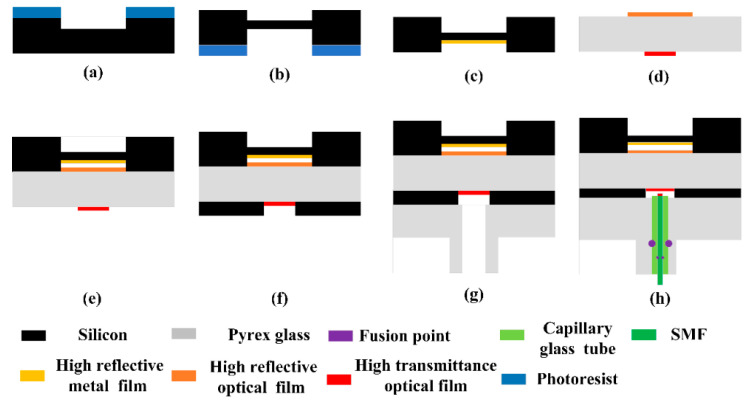
The manufacturing process of a high-temperature fiber-optic Fabry–Perot pressure sensor: (**a**) deep silicon etching on the silicon surface of Fabry-Perot cavity., (**b**) deep silicon etching on the silicon surface of the pressure-sensitive surface., (**c**) plating metal film on the silicon surface in the Fabry-Perot cavity, (**d**) plating dielectric films on two surfaces of Pyrex glass, (**e**) anodic bonding between the sensitive silicon diaphragm and the glass back plate, (**f**) anodic bonding between the glass back plate and the through-hole silicon, (**g**) anodic bonding between the through-hole silicon and the glass cover plate, and (**h**) the capillary tube with optical fiber and the sensitive unit are welded.

**Figure 5 micromachines-13-00763-f005:**
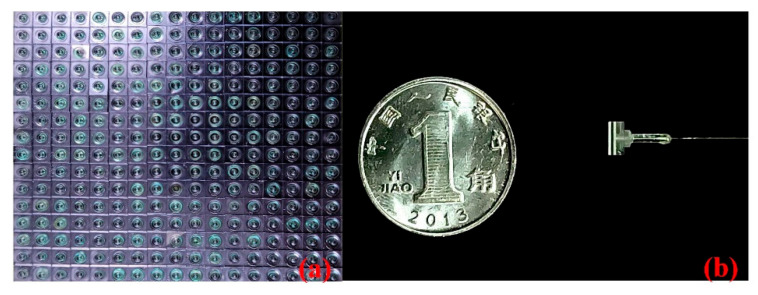
The diagram of finished sensors: (**a**) mass-produced sensitive units of the sensor, and (**b**) the fiber-optic Fabry–Perot sensor.

**Figure 6 micromachines-13-00763-f006:**
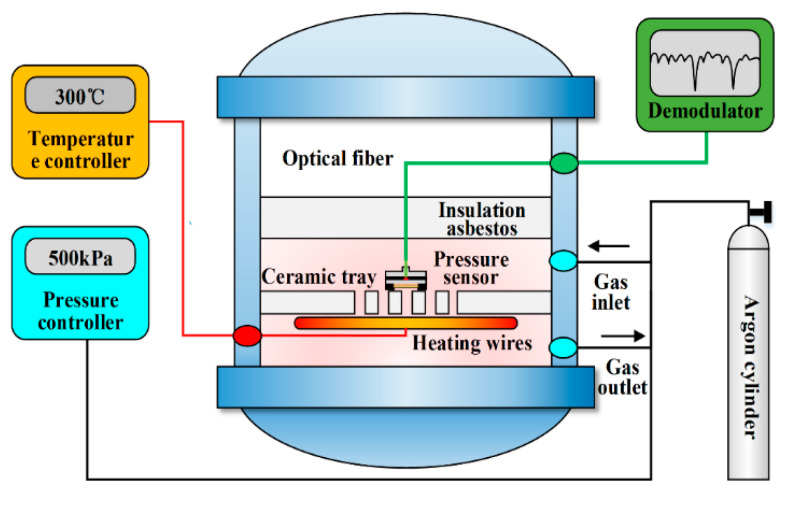
Schematic diagram of the experimental setup.

**Figure 7 micromachines-13-00763-f007:**
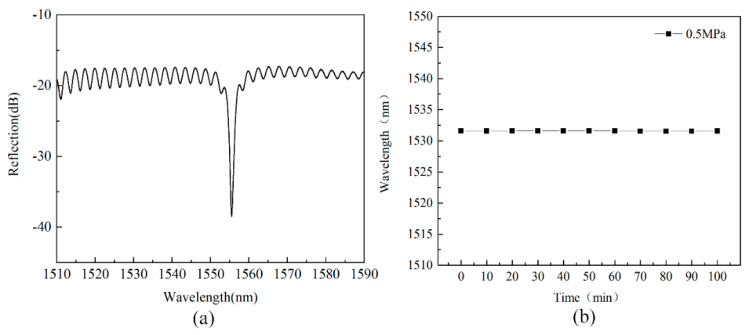
The output of the fiber-optic Fabry–Perot pressure sensor: (**a**) the multi-beam spectrum of the sensor at room temperature, and (**b**) the wavelength of the spike under 0.5 MPa constant pressure.

**Figure 8 micromachines-13-00763-f008:**
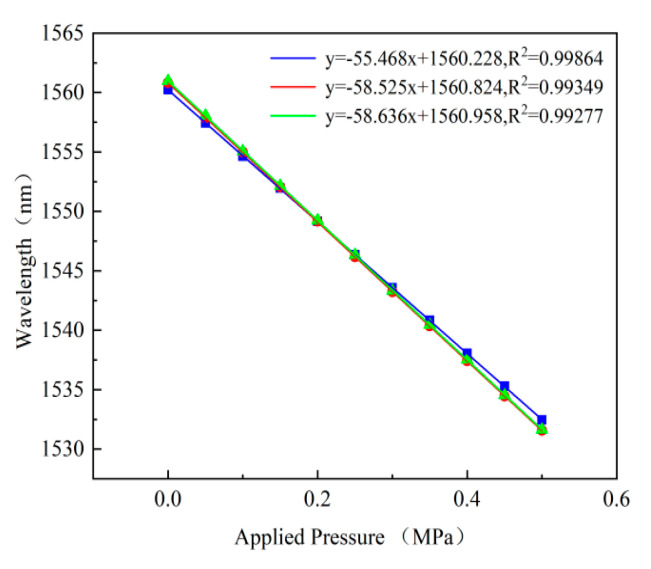
The repeatability of the sensor.

**Figure 9 micromachines-13-00763-f009:**
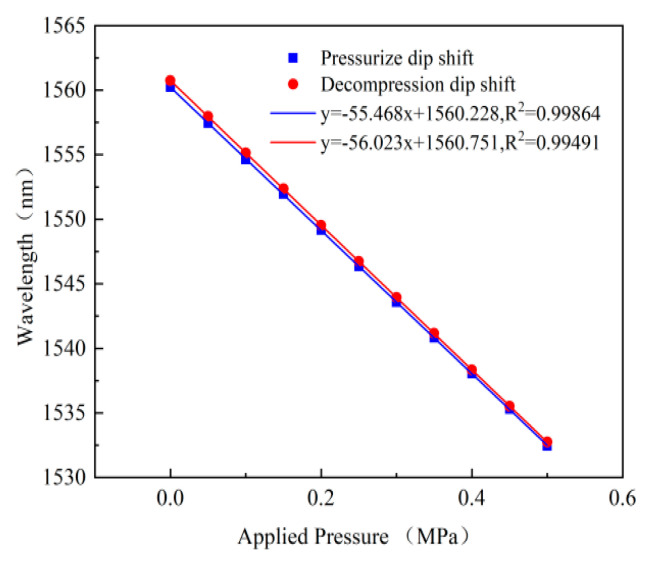
The relationship between the wavelength and the applied pressure.

**Figure 10 micromachines-13-00763-f010:**
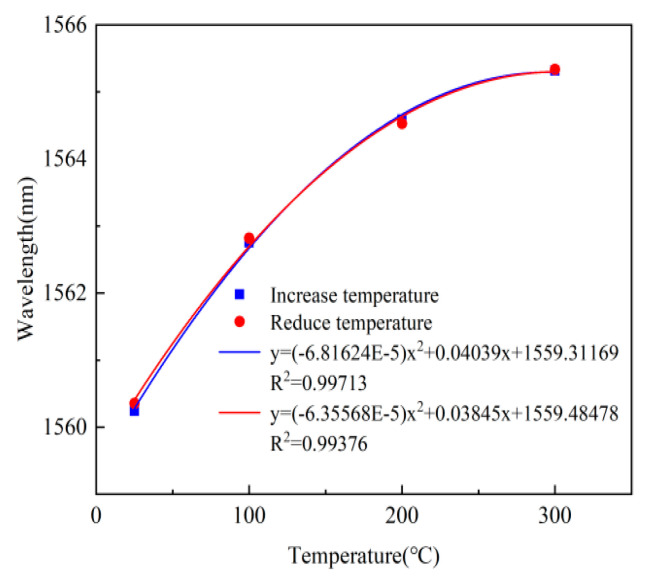
The relationship between the wavelength and the operating temperature.

**Figure 11 micromachines-13-00763-f011:**
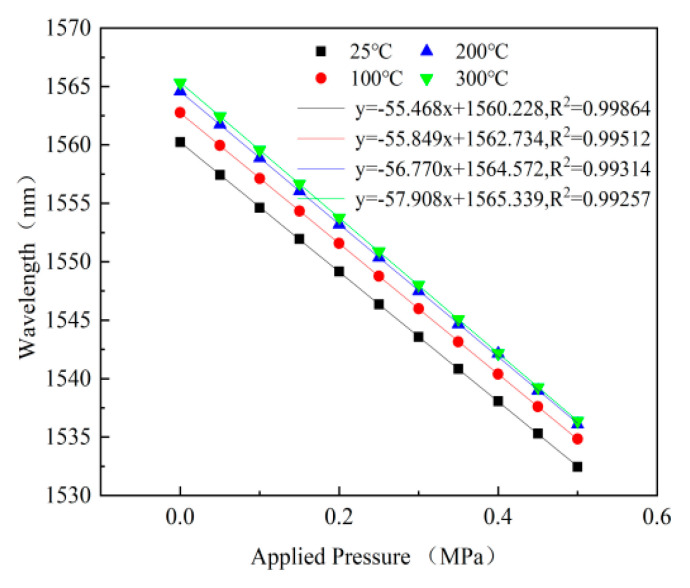
The relationship between peak wavelength and pressure at 25, 100, 200 and 300 °C.

**Table 1 micromachines-13-00763-t001:** The parameters of fiber-optic Fabry–Perot pressure sensor.

Parameters	Symbol	Units	Value
Radius of diaphragm	R	mm	0.78
Thickness of diaphragm	h	μm	100
Young’s modulus of sensitive diaphragm	E	GPa	129.5
Poisson’s ratio of sensitive diaphragm	μ		0.278
Pressure range	P	MPa	0.5
Initial length of FP cavity	L	μm	30

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
