# Peer review of "A MEMS-Based High-Fineness Fiber-Optic Fabry–Perot Pressure Sensor for High-Temperature Application"

_micromachines, 2022, doi:10.3390/mi13050763_

Round 1

Reviewer 1 Report

This article shows the fabrication and the performance of the Fabry-Perot pressure sensor by MEMS techniques.

The fabrication process is well-organized, and the performance of the sensor seem is available for practical use.

Please consider to improve the manuscript at several points as follows:

(1) In sec.3, please describe the manufacturing dimensions for wafer process more precisely. cf, which size of wafer did authors use? How thick is the pyrex glass?  Pitch size among the devices?

(2) Please describe materials precisely for “high reflective metal film” and “high reflective optical film” in Fig.4.

(3) In L.124, Figure 4j -> Figure 4(h).

(4) Fig.9 should be referred in the main body of the manuscript. (L.166-174?)

(5) How is the response of this F.P. pressure sensor system? Did the wavelength value react immediately after changing circumstances such as temperature, pressure etc?

(6) How about the hermeticity and reliability of this sensor? Is there any cycle test results?

Reviewer 2 Report

This paper reports an  high-fineness optical fiber FP high-temperature pressure sensor based on MEMS technology. I have some comments.

  1. When is mentioned: "There are many different types of pressure sensor have been developed in recent years, including piezoresistive [10], piezoelectric [11], capacitive [12] and fiber-optic sensors [13]" - I miss  additional references for fiberoptics sensor since this ref. 13 is not the best one for pressure sensor and also needs to be highlighted here that the pressure sensor sometimes is required to be independent of temperature or at least to be a simultaneous measurement. Please read and add: Optics & Laser Technology 131, 106440, 2020; Materials Letters 271, 127810, 2020; Optics & Laser Technology 112, 77-84, 2019.  It was ignored a lot of critical literature about pressure sensors.

2. Why so many probes used (Fig. 5)? Fig. 5b is not well clear to see the sensor. Please improve.

3. How was optimized the parameters from Table1?

4. Please show some FP spectra of different probes produced. How identical they are to guarantee the reproducibility on the production? Is it  easy to get similar spectrum? Please comment.

5. How can the humidity be influence the sensor performance?

Round 2

Reviewer 2 Report

I am happy with the revision.